# Development and Validation of a Virtual Reality Simulator for Robot-Assisted Minimally Invasive Liver Surgery Training

**DOI:** 10.3390/jcm11144145

**Published:** 2022-07-17

**Authors:** Alan Kawarai Lefor, Saúl Alexis Heredia Pérez, Atsushi Shimizu, Hung-Ching Lin, Jan Witowski, Mamoru Mitsuishi

**Affiliations:** 1Department of Bioengineering, School of Engineering, The University of Tokyo, Tokyo 113-8656, Japan; qlin1806@g.ecc.u-tokyo.ac.jp (H.-C.L.); mamoru@g.ecc.u-tokyo.ac.jp (M.M.); 2Department of Surgery, Jichi Medical University, Tochigi 329-0498, Japan; ashimizu@jichi.ac.jp; 3Research and Development Center, Sony Group Corporation, Tokyo 108-0075, Japan; alexishprez91@gmail.com; 4Department of Radiology, NYU Grossman School of Medicine, New York, NY 10016, USA; jwitos@gmail.com; 5Department of Mechanical Engineering, School of Engineering, The University of Tokyo, Tokyo 113-8656, Japan

**Keywords:** simulation training, kinematic parameters, virtual reality

## Abstract

The value of kinematic data for skill assessment is being investigated. This is the first virtual reality simulator developed for liver surgery. This simulator was coded in C++ using PhysX and FleX with a novel cutting algorithm and used a patient data-derived model and two instruments functioning as ultrasonic shears. The simulator was evaluated by nine expert surgeons and nine surgical novices. Each participant performed a simulated metastasectomy after training. Kinematic data were collected for the instrument position. Each participant completed a survey. The expert participants had a mean age of 47 years and 9/9 were certified in surgery. Novices had a mean age of 30 years and 0/9 were certified surgeons. The mean path length (novice 0.76 ± 0.20 m vs. expert 0.46 ± 0.16 m, *p* = 0.008), movements (138 ± 45 vs. 84 ± 32, *p* = 0.043) and time (174 ± 44 s vs. 102 ± 42 s, *p* = 0.004) were significantly different for the two participant groups. There were no significant differences in activating the instrument (107 ± 25 vs. 109 ± 53). Participants considered the simulator realistic (6.5/7) (face validity), appropriate for education (5/7) (content validity) with an effective interface (6/7), consistent motion (5/7) and realistic soft tissue behavior (5/7). This study showed that the simulator differentiates between experts and novices. Simulation may be an effective way to obtain kinematic data.

## 1. Introduction

Simulation has played an increasingly prominent role in surgical education since the coalescence of three major changes in education, including the advent of laparoscopic surgery (1987), the Institute of Medicine report (1999), and duty hours restrictions (2003). These three events demanded changes in clinical practice as well as surgical education. There have been studies of the educational value of simulation education using simulators with a wide range of designs [1].

At the present time, the clinical benefits of using robot-assisted surgery have not been proven, especially compared to laparoscopic surgery, despite the increased cost and operative time associated with robot-assisted surgery [2]. However, the widespread use of robot-assisted surgery, demands changes in education. Simulation is valuable for teaching manual skills in a safe environment, and objective measures of surgical performance are also an important goal.

Since robot-assisted surgery is indirectly controlled by the surgeon’s hand, there is an opportunity to query the control computer for pose data. The da Vinci Surgical System for robot-assisted surgery (Intuitive Corp, Sunnyvale, CA, USA) collects kinematic data at frequencies up to 100 Hz, including 192 data values at each time point [3]. The kinematic data collected by the da Vinci robot include detailed position information about the two hand controllers and the two instrument tips, such as their position in x, y, and z space as well as their rotational position. However, access to the data is extremely limited. Studies of hand motions during robot-assisted surgery are needed but are difficult to conduct.

Simulation, especially to teach laparoscopic surgery, has been developed with a wide range of equipment ranging from cardboard boxes to computer-based virtual reality simulators. There have been studies comparing video games to surgery simulators [4,5]. One of the key features that makes video games successful is accessibility, which may also be important in the design of surgery simulators [6]. Virtual reality for surgical simulation has advanced based on principles used in the video game industry and has been made possible due to rapid advances in computer technology, including processor speed and inexpensive memory. Virtual reality training has been used extensively for training in laparoscopic surgery. In three trials comparing virtual reality simulation training to box or video trainers in a review, training was considered equal with either type of simulator [1].

A recent Delphi consensus conference was convened to discuss the importance of virtual reality simulation for education in robot-assisted hepatic surgery [7]. Despite this perceived need, there are no readily available virtual reality simulators dedicated to training in hepatic surgery. Most virtual reality simulators available today are used for teaching specific surgical skills such as suturing and general skills such as peg transfer [8,9,10]. There are currently four platforms that are widely used for robot-assisted surgery simulation [10]. The total cost of a functional system using these four platforms varies from USD 100,000 to USD 600,000 [10]. Assessment for all four systems uses different proprietary software to calculate a variety of metrics and there does not appear to be any way to directly access kinematic data [10].

This study was undertaken to develop a virtual reality simulator for training in robot-assisted liver surgery that would provide kinematic data for further analysis and development.

## 2. Materials and Methods

### 2.1. Simulator Design

For simulation efficiency, a hybrid software model was selected, and the simulator was coded in C++ using PhysX (NVIDIA, Santa Clara, CA, USA), an impulse-based simulator for rigid body dynamics [11] and FLEX (NVIDIA, Santa Clara, CA, USA) a position-based dynamics simulator [12] to support soft body deformation. A single hand-held X-box controller (Microsoft Corp, Redmond, WA, USA) is used to direct one instrument at a time through the x, y and z axes with two joystick controls and enables activation of the cutting function with one button. The simulator creates a file of kinematic data at each use in the same format as the JIGSAWS database [13]. The overall design of the simulator is shown in Figure 1. The simulator functioned on a standard personal computer with an NVIDIA graphics processing unit and provided video at or above 60 frames per second (fps). In addition, the position of the instrument relative to the tumor when activated and the number of activations was recorded. The liver surgery simulator was designed with two functioning instruments, both with actions similar to ultrasonic shears.

### 2.2. Liver Model

The liver model used was created directly from computed tomography scan data from a patient (Figure 2). A tetrahedral-mesh model of the liver was created using 3D Slicer (www.slicer.org, accessed on 12 August 2019) and Autodesk Maya (San Rafael, CA, USA) [14]. The left hepatic vein is deep in the left medial segment. The left medial segment was modeled with tetrahedra as shown in Figure 2. The right lobe and left lateral segment were static, creating a hybrid model for computing efficiency [15].

### 2.3. Kinematic Data Analysis

Software was developed to produce kinematic data in a format similar to data from the da Vinci Surgical System and that in the JIGSAWS database [3,13,16] and is based on previous studies with ROVIMAS [17]. The software analyzes the data file produced by the simulator and calculates the time for a procedure, the number of movements and path length. These three parameters form the basis of the motion analysis of robot-assisted surgery data. Time is the time in seconds from the start of the procedure until completion. The number of movements is the number of times that the instrument is discretely moved. This value is calculated from the coordinate data collected as kinematic data by the simulator. The path length is the total number of centimeters that the tip of the instrument moves during the procedure.

### 2.4. Validation Study

A validation study of the liver surgery virtual reality simulator was conducted. This study was approved by the Institutional Review Board of Jichi Medical University prior to conducting the study. Study subjects were recruited including 9 surgically naïve operators (novices) and 9 expert surgeons, and all subjects provided written informed consent. The study consisted of performing a virtual reality metastasectomy of a lesion in the left medial segment of the liver with the simulator developed as described above (Figure 3).

### 2.5. Participants

The inclusion criteria for novice surgeons were those with no experience in performing any part or all of any surgical procedure. The inclusion criteria for expert surgeons were those board-certified in general surgery and with no experience of performing robot-assisted surgery of any kind.

### 2.6. Performance Assessment

Assessment was conducted by the analysis of kinematic data provided by the simulator as well as a self-assessment questionnaire completed after the exercise. The questionnaire included five questions regarding face validity, content validity and fidelity of the simulator. All questions used a 7-point Likert scale.

### 2.7. Sample Size Determination

The sample size for the validation study was determined before the study using α = 0.05 and β = 0.20, with estimated values [18] which showed that each group should have 8 study subjects.

### 2.8. Statistical Analysis

Data for time, movements and path length were collected and grouped according to experience level (novice, expert). Data were compared using the Mann–Whitney U-test [19]. A *p*-value of <0.05 was considered significant.

## 3. Results

### 3.1. Participants

The study participants included nine surgical novices (six males, three females, mean age of 30 years old), with a mean of seven laparotomy procedures observed and one liver procedure. There were nine surgical experts (eight males, one female, mean age of 47 years old, 9/9 board certified general surgeons, 3/9 certified in laparoscopic surgery and 2/9 certified in hepatobiliary surgery) with a mean of 21 years of postgraduate experience. The expert surgeons had performed a mean of 650 laparotomies and 90 liver procedures. None of the expert surgeons had performed robot-assisted surgical procedures.

### 3.2. Simulation Exercise

Study participants performed a simulated metastasectomy (Figure 3) using simulated ultrasonic shears after a brief introduction to using the X-Box controller. Software calculated the time, path length, number of movements, number of times the instrument was activated, and percent of instrument activations at less than 2 cm from the edge of the tumor. The robot-assisted surgery skills of the participants were assessed using the Global Evaluative Assessment of Robotic Skills (GEARS) tool, which was specifically designed to assess robot-assisted surgery skills [20]. Evaluations were performed in six domains, each scored with 1–5 points for a maximum score of 30. The GEARS tool has shown excellent consistency, reliability and validity. GEARS scores were recorded for all participants by a single volunteer observer (board-certified surgeon and full-time surgical educator). The results of the data analysis for nine novice and nine expert participants are shown in Table 1. There are significant differences between these two groups with regard to time, path length, movements and GEARS scores. There were no significant differences between the groups for the number of times the instrument was activated or percent of activations near the tumor.

### 3.3. Post-Simulation Questionnaire

After the simulation exercise, participants completed a questionnaire with five questions about the simulator scored on a 7-point Likert scale. Participants considered the simulator realistic (median 6.5/7) (face validity), appropriate for education (5/7) (content validity) with an effective interface (6/7), consistent motion (5/7) and realistic soft tissue behavior (5/7). The questionnaire (translated to English) is shown in Table 2.

## 4. Discussion

This liver surgery simulator was developed in keeping with several basic principles. First, there is a need for virtual reality simulators to train in liver surgery, and there is a paucity of such simulators. Second, while kinematic data collected during robot-assisted surgery may be useful for the objective analysis of surgical skill, obtaining this data is limited and we sought a simulator that provided kinematic data for further analysis using a standard format. Third, the simulator should provide a good user experience with smooth video and be easily accessible to surgical trainees without the need for highly specialized computer hardware.

The software design approach selected for this liver simulator uses a combination of existing software modules and new code such as NVIDIA PhysX and NVIDIA Flex. Along with the software design, the management of the physics by the simulator must be considered, that is, how the simulator will handle collisions between the simulated components. Position-based dynamics are considered to be fast and stable and model position displacements directly, although they are not as accurate as the finite element model [21]. Position-based dynamics are used by NVIDIA FleX [12,22]. NVIDIA FleX was employed for soft body deformations and uses a particle-based model for a wide variety of objects.

The liver model employed here is a tetrahedral mesh that was created from patient data. The mesh is composed of particles organized as tetrahedra. Simulation speed is greatly affected by the number of particles in the model. For this reason, only the left medial segment of the liver was modeled as a mesh that can be modified (Figure 2). Modeling the entire liver would degrade the performance of the simulator to an unacceptably low level. The simulation of the instrument function was an important characteristic in the overall design of the liver surgery simulator. The choice of user interface was important for this simulator. For this study, we selected the X-box controller since it is well-known to many people and is inexpensive, thus making the simulation system more accessible.

The liver was presented as a hybrid model [16] with the right lobe and left lateral segments as fixed structures and the left medial segment as a tetrahedral mesh, which can be acted upon by the instruments. Participants judged that the instrument motion was smooth and the soft tissue behavior was realistic in the post-simulation questionnaire. The simulator is accessible, runs on a standard PC using an inexpensive controller with high quality video at 60 fps, and should be useable by trainees in the hospital or at home.

The simulation exercise selected was the metastasectomy of a lesion on the edge of the left medial segment. The simulator is anatomically accurate for this procedure or for a left lateral segmentectomy. Metastasectomy was chosen because it is relatively simple and thus suitable for expert surgeons as well as surgical novices. The simulator used here does not include bleeding from the liver parenchyma, but the left hepatic vein is included and will bleed if cut (Figure 4). In discussions with expert surgeons, we concluded that parenchymal bleeding is not needed and would only distract junior surgeon performing a simulated metastasectomy.

Magnetic sensors attached to the surgeons’ hands were used to obtain motion data during laparoscopic cholecystectomy and investigators have concluded that time, path length and movements are valid parameters to differentiate operators by skill level [23]. The availability of motion data in robot-assisted surgery is a major paradigm shift in this field; however, access to the data is very limited. Time and path length have been shown to have construct validity in several studies [17,23,24,25] and the number of movements has also been validated [17,23,24]. These three parameters have been used in previous studies of robot surgery to assess surgical performance. In the present study, data for the position of the instrument tip from the simulator are used to calculate time, path length and movements. The calculation of these parameters from kinematic data has been described previously and is considered to be established as a standard method [17,23].

Data from the present study showed that these three parameters can differentiate novice from expert surgeons using this liver simulator with a metastasectomy exercise. After developing the simulator, we conducted a study to validate the simulator. There are three types of validity to address: face validity (realism of the simulator), content validity (appropriate as a teaching modality) and construct validity (distinguish experienced from inexperienced users) [26]. In this study, face validity and content validity were verified by a user survey after the exercise. Time, path length and movements have been shown to be valid parameters for surgical skill assessment and were used in this study [24,25].

There were significant differences in all three parameters for the two groups of participants, which is consistent with construct validity. It is notable that responses to the survey for the two groups (novices, experts) regarding their opinion about the simulator (validity and fidelity) were not significantly different for the two participant groups; a similar result was also previously reported [27].

Simulation training for robot-assisted surgery has not been studied extensively yet. It is not clear which metrics are the most important. In a study of four commercially available robot-assisted surgery simulators, investigators concluded that virtual reality-based training does improve basic surgical skills [9]. However, the key issue is to determine how training in those skills transfers to clinical surgery. To answer that question, a study of expert surgeons compared the robot-assisted simulator performance using skill exercises with reviews of robot-assisted surgical videos and found no correlation [28].

Previous studies of kinematic data analysis with simulators have been based on proprietary hardware. This new liver surgery simulator is software-based without the need for proprietary hardware and is thus more accessible to trainees. Although this data is not routinely available to investigators who use the da Vinci system, appropriately designed simulators offer a viable alternative for training and the acquisition of kinematic data. This is the first liver surgery procedural simulator and it provides kinematic data in a standard format to facilitate further analysis and development. Further development of this simulator will continue and may allow interchangeable instruments including stapling devices, and the ability to expand the simulated liver to the right lobe. Future versions of the software may facilitate these capabilities.

There is a need for accessible virtual reality liver surgery simulators to advance training in liver surgery. The use of kinematic data for skill assessment is hampered by the lack of available data. The simulator developed here uses readily available hardware, provides standard kinematic data, and can differentiate novice and expert surgeons based on significant differences in time, movements and path length using data obtained during a simulated hepatic metastasectomy, as well as GEARS scores. Simulation may be an effective way to obtain kinematic data for further analysis and development.

## Figures and Tables

**Figure 1 jcm-11-04145-f001:**
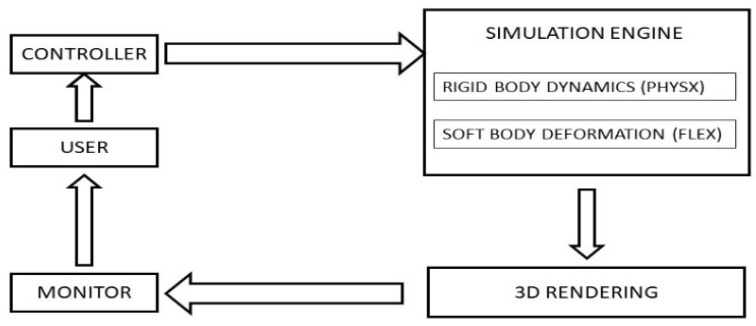
Simulator system diagram.

**Figure 2 jcm-11-04145-f002:**
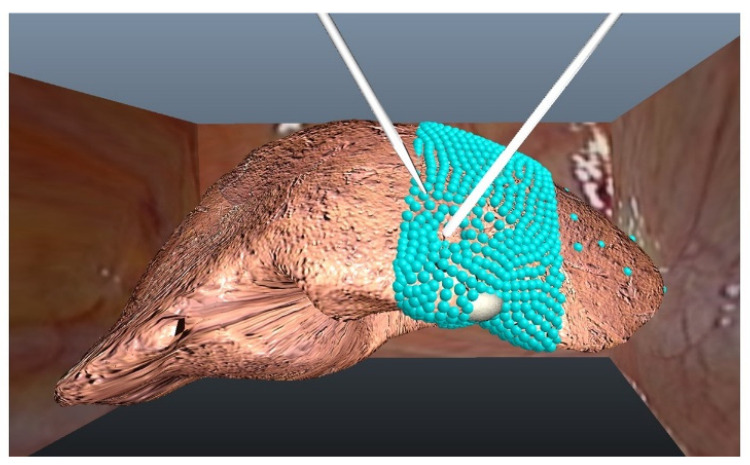
Liver model. This view shows the entire liver as modeled, with the particles in the tetrahedral mesh visible (they are not shown during the simulation). The single metastatic lesion is at the inferior edge of the left medial segment shown as a solid sphere. The area shown as particles is deformable when acted upon by either of the two instruments shown.

**Figure 3 jcm-11-04145-f003:**
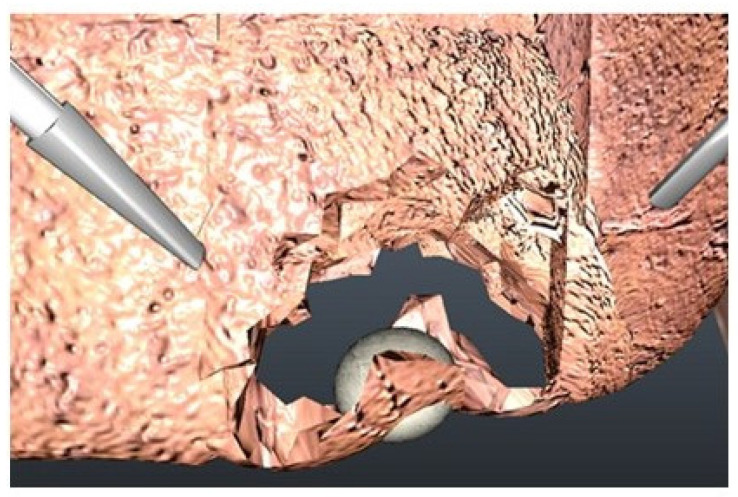
Simulated metastasectomy.

**Figure 4 jcm-11-04145-f004:**
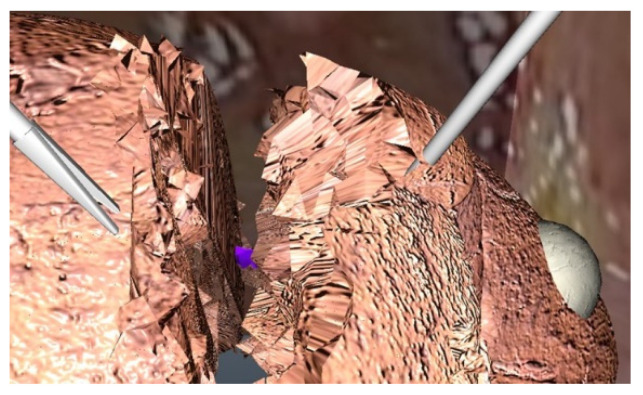
Left hepatic vein as seen during transection of the left lateral segment.

**Table 1 jcm-11-04145-t001:** Liver surgery simulator: assessment parameters for novice and expert surgeons in a validation study.

Participant	Time (s)	Movements	Path Length (m)	GEARS Score	Instrument Activations	Percent Close Instrument Activations (<2 cm)
Novice N = 9	174 ± 44	138 ± 45	0.76 ± 0.20	19.2	107.4	24.8
Expert N = 9	102 ± 42	84 ± 32	0.46 ± 0.16	26.7	108.7	22.9
*p*-Value	*p* = 0.004	*p* = 0.043	*p* = 0.008	*p* = 0.001	*p* > 0.05	*p* > 0.05

**Table 2 jcm-11-04145-t002:** Post-simulation questionnaire.

1. The VR liver simulator is sufficiently realistic.						
1	2	3	4	5	6	7
Disagree		No opinion		Strongly Agree		
2. The Virtual Reality liver simulator is an appropriate modality for surgical training.						
1	2	3	4	5	6	7
Disagree		No opinion		Strongly Agree		
3. The VR liver simulator has an effective interface for training junior residents and students.						
1	2	3	4	5	6	7
Disagree		No opinion		Strongly Agree		
4. The instruments move consistently during the procedure.						
1	2	3	4	5	6	7
Disagree		No opinion		Strongly Agree		
5. The soft tissue behaves in a realistic manner when using the simulator.						
1	2	3	4	5	6	7
Disagree		No opinion		Strongly Agree		

## Data Availability

All data collected in this study are reported. The original data can be obtained from the corresponding author upon request.

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
