# Peer review of "Development and Validation of a Virtual Reality Simulator for Robot-Assisted Minimally Invasive Liver Surgery Training"

_jcm, 2022, doi:10.3390/jcm11144145_

Round 1
Reviewer 1 Report
The aim of the study aimed to describe efficacy of virtual reality simulator for training and expert surgeons. Unfortunately, this study suffers from many limitations.
First, the sample size is to small to perform any analysis, this is the main limitation of the study. Description of inclusion criteria must be included, as well as definition of novice and expert surgeon, based on case load. Parameters defined to evaluate efficacy of simulator must be described, and should be discuss why authors selected these parameters as significant for comparison between groups. Baseline clinical features are missing, in order to describe baseline differences between groups.
Reviewer 2 Report
Thank you for asking me to review the paper entitled "Development and validation of a virtual reality simulator for robot-assisted minimally invasive liver surgery training".
As a clinician, I found this article very interesting and important for further surgical training.
I would suggest several minor changes:
-introduction, lines 29-33, please change the order of three major events, based on the year : 1987 first, followed by 1999 and 2003.
-please introduce the word kinematic data in line 43 on page 1
-Sentence : Some mathematical notation is needed to define these three parameters ..." is not clear. Does it mean that the authors had to do "some" mathematical notation? Please explain more details what does "some" means .
-Page 4, line 118-121. Please add the questionnaire that you used
-Line 143: observed by an observer. Please add more details about the observer: qualification, how she/he was chosen
-Please introduce GEARS scores
-In discussion, there are elements of repetition
-The authors should add their opinion how they think that this method can improve in the future.
Round 2
Reviewer 1 Report
No further comments. Authors properly addressed reviewers comments.